# Johnson-noise-limited cancellation-free microwave impedance microscopy with monolithic silicon cantilever probes

Jun-Yi Shan [1,2], Nathaniel Morrison[1,2], Su-Di Chen [1,2,3], Feng Wang [1,2,3] & Eric Y. Ma [1,2,4] ✉

Microwave impedance microscopy (MIM) is an emerging scanning probe technique for nanoscale complex permittivity mapping and has made significant impacts in diverse fields. To date, the most significant hurdles that limit its widespread use are the requirements of specialized microwave probes and high-precision cancellation circuits. Here, we show that forgoing both elements not only is feasible but also enhances performance. Using monolithic silicon cantilever probes and a cancellation-free architecture, we demonstrate Johnson-noise-limited, drift-free MIM operation with 15 nm spatial resolution, minimal topography crosstalk, and an unprecedented sensitivity of 0.26 zF/√Hz. We accomplish this by taking advantage of the high mechanical resonant frequency and spatial resolution of silicon probes, the inherent common-mode phase noise rejection of self-referenced homodyne detection, and the exceptional stability of the streamlined architecture. Our approach makes MIM drastically more accessible and paves the way for advanced operation modes as well as integration with complementary techniques.

Microwave impedance microscopy (MIM) allows nanoscale mapping of local complex permittivity in a wide variety of systems under diverse conditions[1–5], from buried 2D electronic systems at mK temperatures and topological materials in strong magnetic fields, to semiconductor devices under optical illumination and biological samples in liquid[6–13]. MIM achieves this by delivering microwave fields to a sharp metallic tip and measuring the minuscule changes in the complex reflection coefficient due to near-field tip-sample interaction.

To date, two elements are believed to be essential to MIM: specialized probes for microwave delivery that minimize stray coupling, probe capacitance, and ohmic loss; and a high-precision cancellation circuit that removes the baseline portion of the reflected microwave via destructive interference, thereby allowing high gain to achieve the required sensitivity (Fig. 1a).

Nevertheless, these two elements introduce significant complexity, raising barriers to broader adoption and, more importantly,

compromising the performance limits of MIM. The specialized microwave probes—ranging from stripline cantilevers requiring intricate micro-electromechanical fabrication[14] to manually etched and glued long-shank bare metal wires and cantilevers[15,16]—typically provide only moderate tip sizes with low reliability and reproducibility compared with monolithic silicon probes. Furthermore, they exhibit relatively low mechanical resonant frequencies and quality factors (Q-factors). Such limitations constrain the ultimate topography performance, introduce artifacts, and hinder consistent, quantitative interpretation of MIM signals.

Furthermore, the cancellation circuits, typically comprising multiple stages of mechanical and voltage- or digitally-controlled amplitude attenuators and phase shifters, are prone to external disturbance. Vibrations and electromagnetic interference can inject noise directly into the un-amplified MIM signal. Ambient temperature drifts and device aging lead to a fluctuating baseline that ultimately saturates the

[1]Department of Physics, University of California, Berkeley, Berkeley, CA 94720, USA. [2]Lawrence Berkeley National Laboratory, Berkeley, CA 94720, USA. [3]Kavli Energy NanoScience Institute, University of California, Berkeley, Berkeley, CA 94720, USA. [4]Department of Electrical Engineering and Computer Sciences, University of California, Berkeley, Berkeley, CA 94720, USA. ✉e-mail: eric.y.ma@berkeley.edu

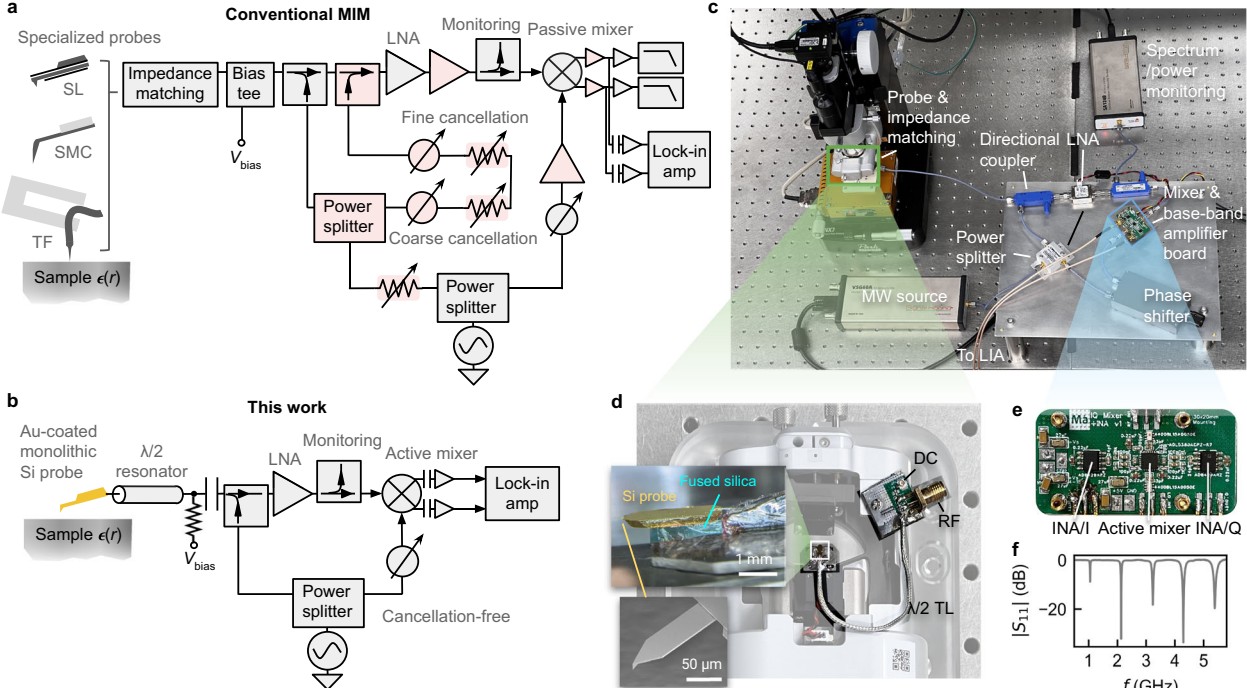

**Fig. 1 | Cancellation-free MIM with monolithic silicon cantilever probes.** Design of conventional MIM (**a**) and this work (**b**). Components removed in our architecture are shaded red. LNA low-noise microwave amplifier, SL stripline, SMC solid metal cantilever, TF tuning fork. **c** Implementation of our design. MW microwave, LIA lock-in amplifier. **d** Details of the probe and impedance matching network attached to the atomic force microscope (AFM) head. DC direct current, RF radio frequency. Inset probe mounting details and scanning electron micrograph of the Si probe[25]. **e** The custom printed circuit board integrating the active mixer and INAs. **f** $|S_{11}|$ spectrum of the Si probe impedance matched by a half-wave resonator. Source data are provided as a Source Data file.

amplifier chain, limiting integration times and thereby restricting achievable sensitivity. Additionally, the cancellation circuits often contain bandwidth-limiting components.

In this work, we present a Johnson-noise-limited, drift-free MIM with 15 nm electrical spatial resolution, minimal topography crosstalk, and a sensitivity of 0.26 zF/$\sqrt{Hz}$ using monolithic silicon probes and a cancellation-free architecture. We describe how we bypass the need for specialized probes or cancellation, and in turn, advance the state of the art in MIM performance. We discuss how our approach facilitates advanced MIM operation such as spectroscopy and nonlinear mode, and seamless integration with complementary techniques that utilize high-performance silicon-based probes such as near-field optical, magnetic force, and acoustic force microscopy[17–19].

## Results
### Design and validation
We first briefly review the principle and design considerations of conventional MIM (Fig. 1a). A source generates microwave which is split into three branches. The first branch is delivered to specialized MIM probes through a directional coupler and an impedance-matching network[20,21]. The second branch passes through multiple stages of tunable attenuators and phase shifters and is combined with the microwave reflected from the tip sample in order to cancel the baseline reflection through destructive interference. The baseline-free signal is then amplified by several low-noise amplifiers (LNA), and mixed with the third branch to down-convert the signal to baseband frequencies. An additional directional coupler is often inserted between the LNA and the mixer for monitoring. The resulting in-phase (I) and quadrature (Q) signals are then amplified by baseband amplifiers and detected after a low-pass filter, or, in dynamic mode, through a lock-in amplifier (LIA) locked to the probe's mechanical vibration or other modulations.

Our design differs from the above in several important aspects, with the most significant being the adoption of standard gold-coated monolithic silicon cantilever probes, and the complete elimination of the cancellation circuits (Fig. 1b, c). In addition, we opted for a single LNA and an active mixer and directly AC-coupled the I and Q outputs before further amplification through high-gain instrumentation amplifiers (INA). We integrated the active mixer and INAs onto a single custom printed circuit board (PCB) to minimize signal loss and noise (Fig. 1e). See Methods section "MIM electronics and probes" for more details.

Defying the conventional wisdom, our MIM implementation is not only operational but also high-performing. For an initial evaluation, we used an Al dot sample, featuring $5\,\mu m \times 5\,\mu m \times 15$ nm Al squares patterned on SiO$_2$. Figure 2 presents a side-by-side comparison of images taken with our approach and with a state-of-the-art commercial MIM system using multi-stage cancellation and a solid metal cantilever (SMC) probe, at identical microwave power and scan rate. Both sets correctly show strong contrasts between Al and SiO$_2$ in the reactive channel (MIM-Im) and minimal contrast in the dissipative channel (MIM-Re)[2,22]. The lower MIM-Im signal associated with surface particulates on Al proves that the MIM signals originated from local conductivity instead of topography crosstalk. The same dot was imaged, and the sharper Si tip gave rise to a more faithful representation of the surface features. Despite a higher spatial resolution, which generally leads to trade-offs in MIM signal, our implementation shows an excellent signal-to-noise ratio (SNR), as evidenced by the scan line profiles (Fig. 2h).

Below we first examine the factors behind this surprising result, addressing the viability of Si probes and the implications of eliminating cancellation, and then show how our approach elevates the state of the art in noise, sensitivity, and spatial resolution.

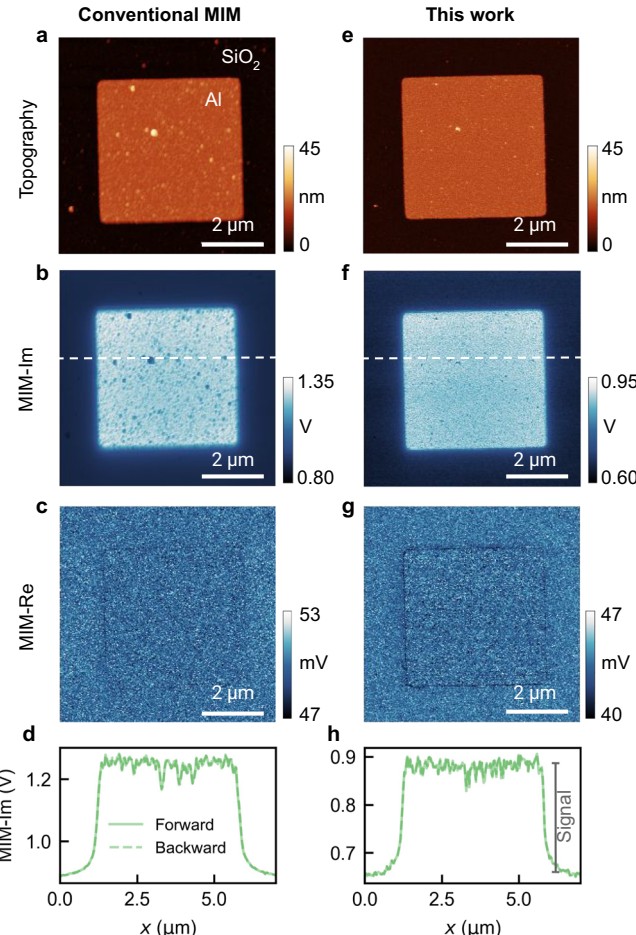

**Fig. 2 | Validation and comparison with state-of-the-art conventional system.** Al dot on $SiO_2$ measured by conventional MIM: (**a**) surface topography, (**b**) raw MIM-Im image, (**c**) raw MIM-Re image, and (**d**) line profiles along the horizontal line in (**b**) showing forward and backward scans. **e–h** The same set of measurements with our implementation. For both sets, the microwave frequency is 3.2 GHz, $P_{in} = -10$ dBm, and the lock-in time constant is 4 ms. Source data are provided as a Source Data file.

## Monolithic silicon probes for microwave imaging

Metal-coated monolithic silicon cantilever probes, despite their excellent force sensitivity, spatial resolution, robustness, reproducibility, and versatility, have generally been considered unsuitable for microwave scanning probe techniques due to perceived challenges associated with strong stray coupling, large self-capacitance, and high ohmic loss[14,23].

We show that these perceived limitations are largely over-estimated or can be substantially mitigated, and are thus outweighed by the benefits. First, the utilization of a dynamic mode, wherein the MIM signal is modulated by the nm-amplitude mechanical vibration of the cantilever and discerned by a LIA, effectively nullifies stray coupling between the probe body and the sample[24], as corroborated by the high-resolution drift-free MIM images with minimal topographic interference (Fig. 2).

Second, our modeling and characterizations show that the GHz-range dissipation and capacitance of gold-coated Si probes[25] are moderate relative to their Pt SMC or Al stripline (SL) counterparts ($\sim 2\,\Omega/0.1$ pF vs. $0.2\,\Omega/0.03$ pF or $5\,\Omega/1$ pF, see Methods). Moreover, strategic placement of a ground plane transforms the probe body into a short microstrip waveguide, where self-capacitance is compensated by self-inductance to render a characteristic impedance close to $50\,\Omega$ (Fig. 1d inset, see also Methods). Consequently, the Si probes can be reliably impedance matched by, e.g., a simple half-wave transmission

line resonator (Fig. 1d, f) to obtain a responsivity ($\eta = V_{tip}/V_{in}$)[20,21] on par with SMC probes (Fig. 2). This allows us to unlock key benefits of Si probes, such as high mechanical resonant frequency, robust topography feedback, thermally-limited mechanical noise, and compatibility with other complementary techniques.

## Cancellation-free operation

We now discuss why cancellation-free operation is possible. The first challenge is that the baseline reflection, as large as $\sim -20$ dBm, could easily saturate the signal chain that requires $\sim 100$ dB of total gain to detect the minuscule signal. We overcome this by reducing microwave gain, operating in dynamic mode, and removing the baseline after down-conversion. More specifically, we use a single LNA before the mixer and directly AC-couple the down-converted mixer outputs. This way, the DC component (<1 kHz) that represents the baseline is rejected, while AC components at the cantilever vibration frequency $f_0$, representing the dynamic-mode MIM signal, are allowed to pass and amplified further by high-gain (66 dB) INAs (Fig. 1b). The signal chain never saturates as long as the reflected power is less than $\sim -20$ dBm, a condition almost always satisfied with proper impedance matching.

Phase noise from the baseline reflection is another potential challenge, and could seemingly overwhelm the dynamic MIM signal regardless of whether the DC component is removed. For instance, our standard microwave source has a phase noise figure of −115 dBc/Hz at 100 kHz offset from 3 GHz, which would imply a voltage noise density of $-44$ dBV/$\sqrt{Hz}$ near $f_0 = 230$ kHz in the down-converted baseband signal ($P_{in} = -5$ dBm, $|S_{11}| = -18$ dB, total system gain = 110 dB, yellow dashed curve in Fig. 3a). Such noise would overpower all other noise sources and drastically diminish the SNR, precluding cancellation-free operation. Yet, our measured noise density is more than 30 dB lower (yellow solid curve), making high-SNR MIM feasible without cancellation.

This unexpected outcome stems from the intrinsic common-mode phase noise rejection of self-referenced homodyne detection[26]. As the signal and the reference are derived from the same source, they contain the same phase noise, which effectively cancels during mixing. For a signal voltage $V_{sig}(t) = V_1 \cos[\omega t + \phi(t)]$ and a reference voltage $V_{ref}(t) = V_2 \cos[\omega t + \phi_0 + \phi(t)]$, their mixing result is $V_{sig}(t) \cdot V_{ref}(t) = \frac{1}{2} V_1 V_2 [\cos(2\omega t + 2\phi(t) + \phi_0) + \cos\phi_0]$, in which the baseband component is independent of the phase noise $\phi(t)$. Experimentally, this common-mode rejection is finite ($\sim 30$ dB in our setup) due to cable length differences, amplifier phase noise, and phase-to-amplitude noise conversion[27]. Nevertheless, it allows additive broadband noise—dominated by Johnson noise, as we will demonstrate—to become the primary contributor at low to moderate power levels.

## Johnson-noise-limited noise floor

We now show that our MIM implementation achieves Johnson-noise-limited noise floor. In MIM measurements, electronic noise arises from two primary sources: noise that scales with the total reflected power such as the phase noise mentioned above, and additive power-independent noise, which includes Johnson-Nyquist noise, amplifier broadband noise, and electromagnetic interference. The input-referenced Johnson noise density is $k_B T$, where $k_B$ is the Boltzmann constant and $T$ is temperature. Figure 3a illustrates that while residual phase noise dominates at input powers above −7 dBm near $f_0 = 230$ kHz, the noise floor converges to a nearly flat profile, not limited by the spectrum analyzer or INA noise, for $P_{in} \leq -10$ dBm. Meanwhile, the dynamic MIM signal at $f_0$ scales linearly with $P_{in}$ down to −25 dBm (Supplementary Fig. 1).

By plotting the gain-normalized noise density at $f_0$ against various $P_{in}$ levels, we confirm Johnson-noise-limited operation for $P_{in} \leq -10$ dBm (Fig. 3c). Within this regime, the Johnson noise near the GHz carrier frequency, emanating from the impedance-matched probe

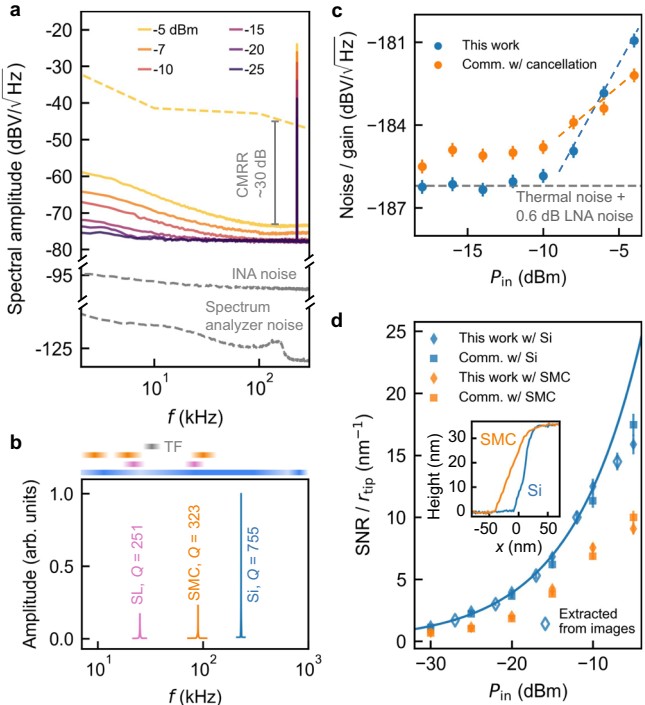

**Fig. 3 | Johnson-noise-limited performance. a** Baseband spectra of MIM-Im signal for various $P_{in}$ at 3.2 GHz for a probe $|S_{11}|$ of −18 dB. The yellow dashed curve shows the estimated phase noise density without common-mode rejection for $P_{in}$ = − 5 dBm. The grey dashed curves show the INA and the spectrum analyzer noise floor. CMRR common-mode rejection ratio. **b** Mechanical vibration amplitude as a function of driving frequency for the stripline (SL), solid metal cantilever (SMC), and Si probes, at the same driving power. The horizontal bars on top indicate the range of available mechanical resonant frequencies $f_0$ for tuning fork (grey), SMC (orange), SL (pink), and Si probes (blue). **c** Gain-normalized noise spectral amplitude near $f_0$ as a function of $P_{in}$. Colored dashed lines are guides to the eye. Gray dashed line shows 295 K Johnson noise plus 0.6 dB LNA noise. **d** Tip-size-normalized SNR as a function of $P_{in}$ for four combinations of MIM electronics and probes. Solid and open symbols are derived from spectra and images, respectively. The solid curve is a fit of the filled blue diamonds at $P_{in}$ < − 10 dBm to $A10^{P_{in}(dBm)/20}$, where $A$ is the fitting parameter. We note that Si probes are not a standard feature of any commercial system. Inset: topography line cuts using two types of probes across a sharp edge to determine the tip radius. Error bars represent the standard error of the mean from four independent measurements. Source data are provided as a Source Data file.

at 295 K is −186.8 dBV/$\sqrt{Hz}$ for 50 Ω and is slightly intensified by the LNA noise (0.6 dB), dominating the measured noise floor after amplification and down-conversion. The LNA's 36 dB gain ensures that noise contributions from downstream components are negligible[28]. To further lower the noise floor, one would need to lower the temperature of both the probe and the LNA.

We attribute this milestone to the high mechanical resonant frequency of Si probes and the significantly streamlined architecture. As shown in Fig. 3a, the maximal power to maintain Johnson-noise-limited operation will decrease with lower probe mechanical resonance frequencies (Fig. 3b) due to elevated phase noise. On the other hand, while cancellation can mitigate phase noise at higher power levels (e.g., for $P_{in}$ > − 6 dBm in Fig. 3c), it inadvertently introduces additional noise. This noise, directly injected into the signal before the initial low-noise amplification, is likely responsible for an observed higher noise floor in the commercial system—up to 2 dB above the Johnson noise, at low and moderate power levels (Fig. 3c).

The Johnson-noise-limited noise floor, coupled with high responsivity, gives rise to excellent performance. Using the SNR normalized by the tip radius as a figure of merit[29], we compare our

approach with various alternate configurations (Fig. 3d). The combination of Si probes and cancellation-free architecture consistently outperforms others in the Johnson-noise-limited regime, where signal increases linearly with $P_{in}$ while noise remains constant (solid curve in Fig. 3d). Moreover, the excellent agreement between SNR values derived from MIM signal spectra and actual images (solid and open diamonds, see also Methods) confirms that the high SNR is achieved in practical scanning with other potential noise sources such as fluctuating tip conditions or vibration amplitudes. This robustness lays the groundwork for achieving the unprecedented sensitivity below.

## Sub-zeptofarad (zF) sensitivity
The excellent tip-radius-normalized SNR, when combined with the exceptional stability of silicon probes and the cancellation-free architecture, allows long pixel dwell times to achieve sub-zF sensitivity while maintaining high spatial resolution. To demonstrate this, we used an encapsulated graphene sample consisting of a monolayer graphene flake sandwiched between layers of hexagonal boron nitride (hBN) (Fig. 4a, b). This configuration highlights our electrical sensitivity with minimal interference from topography: the stark electrical contrast between graphene and hBN takes place across a topographically flat region, whereas the two hBN-only regions show consistent MIM signal despite a 35 nm height difference (Fig. 4c, d).

We then demonstrate an unprecedented sub-zF sensitivity. In principle, sensitivity—defined as the signal corresponding to an SNR of 1—can be enhanced simply by narrowing the measurement bandwidth, thereby reducing the total in-band noise (Fig. 4e). In our dynamic-mode MIM setup, this involves increasing the lock-in time constant and the pixel dwell time to ensure signal stabilization. For instance, we used a lock-in time constant of 4 ms and a pixel dwell time of 16 ms for Fig. 4d, translating to a total in-band noise of 0.17 mV and a sensitivity of 1.7 zF, based on a measured responsivity (see Methods). Practically, however, one is often constrained by cancellation-induced drifts and saturation, as well as challenges like tip degradation and contaminant accumulation, particularly under ambient conditions. Our implementation substantially alleviates these issues, enabling us to further improve the sensitivity.

As an illustrative case, we imaged a 2 $\mu$m × 2.5 $\mu$m area containing two surface particulates (Fig. 4f, g). Using a lock-in time constant of 40 ms and a pixel dwell time of 160 ms, we achieved a sensitivity of 0.53 zF, or a bandwidth normalized sensitivity of 0.26 zF/$\sqrt{Hz}$, representing a 3–4× improvement over previously reported figures in the related scanning capacitance microscopy (SCM) technique[30,31]. We note that MIM significantly differs from SCM due to its simultaneous quadrature sensitivity and minimal sample preparation requirements. The 6 h scan was conducted in an open environment without enclosure or active temperature stabilization—a task inconceivable with conventional MIM. Under identical settings, the commercial system drifted into saturation in less than 2 h (Supplementary Fig. 2). Remarkably, the resulting raw MIM-Im image is virtually drift- and noise-free, has high spatial resolution, reveals detailed permittivity features uncorrelated with topography, and demonstrates excellent consistency between the forward and backward scans (Fig. 4h). This accomplishment shows the possibility of approaching even higher sensitivity, a prospect we explore in subsequent discussions.

## Enhanced spatial resolution through harmonic demodulation
Capitalizing on the robust high-resolution capabilities of silicon probes, we demonstrate additional enhancements in electrical spatial resolution by demodulating the MIM signal at higher harmonics of the cantilever's mechanical resonance (Fig. 5). Contrary to short-ranged interactions like atomic force or tunneling current[32], the near-field microwave interaction is relatively long-ranged due to the power-law decay of electric fields[33]. As a consequence, despite the spatial contrast predominantly originating from the tip apex, contributions from the

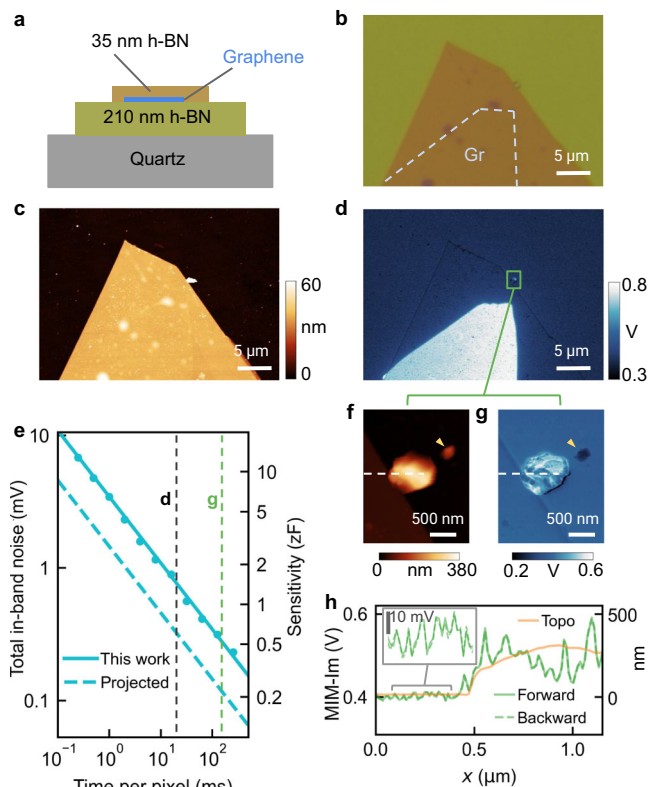

**Fig. 4 | Sub-zF sensitivity. a** Structure of the encapsulated graphene sample. **b** Optical image of the sample. The blue dashed line outlines the graphene region. **c** Topography and (**d**) raw MIM-Im image of the sample. **e** Measured total in-band noise and the corresponding sensitivity as a function of pixel dwell time (circular markers). The sloped lines are theoretical estimates for this work (solid) and a finer proposed measurement at 77 K with lower source phase noise (dashed, see Discussion). The dashed vertical lines indicate the conditions for (**d**) and (**g**). The noise density at room temperature for this work is 0.14 mV/√Hz. **f** A zoomed-in topography image of (**d**). **g** Raw MIM-Im image of the same region taken over 6 h. The smaller feature (yellow arrow) appears insulating with a lower permittivity than hBN and is likely a dust particle; the larger feature appears highly conductive and is likely a mixture of graphene fragments and polymer residue that was pushed out of the top hBN during fabrication. **h** Line profiles from (**f**, **g**) with a zoomed-in view highlighting the SNR. Source data are provided as a Source Data file.

upper sections of the probe can introduce lower-resolution elements, particularly between large regions with high permittivity contrast. This phenomenon is often manifested as "halos" at the boundaries between metallic and insulating regions (Figs. 2, 4, and 5b, e).

We show that demodulating at harmonics of the cantilever mechanical resonant frequency $f_0$ can effectively remove these features, enhancing the electrical spatial resolution and simplifying image interpretation. Taking advantage of the 900 kHz baseband bandwidth, we observe dynamic MIM signals at multiples of $f_0$, albeit with diminishing amplitude (Fig. 5a). Demodulating at higher harmonics provides crisper transitions at the aluminum/SiO₂ boundary (Fig. 5c, d), eliminating the "halo" and achieving tip-radius-limited 15 nm electrical spatial resolution at $3f_0$ while maintaining decent SNR (Fig. 5g inset). The relative contrast from surface contaminants is also enhanced, as the signal becomes increasingly localized.

This technique is effective because it captures the nonlinear dependence of MIM signal to the tip-sample distance that is primarily from interactions near the tip apex. However, this refinement in spatial resolution incurs a reduction in SNR. This trade-off can be mitigated by extending the scan duration, as previously discussed, or increasing the input power, since the harmonics are at higher offset frequencies (Fig. 3a). Notably, while similar methods have been

widely used in other scanning probe techniques[34], demonstrations of such resolution enhancement for MIM imaging have not been previously documented[24].

## Discussion

We have presented an approach to MIM that elevated the state of the art in several aspects. Next, we discuss the promising prospects this work offers, both in enhancing MIM itself and in integrating it with other complementary techniques.

### Spectroscopic mode

Analogous to the leap from scanning tunneling microscopy to spectroscopy, achieving broadband microwave impedance spectroscopy (MIS)—which measures the local impedance spectrum that allows distinguishing between permittivity and conductivity contributions and investigating narrow-band microwave resonances[35]—has been a long-sought goal. However, a major obstacle to continuously tunable broadband operation has been the cancellation circuit, now no longer a necessity. We anticipate that by implementing resistive impedance matching and using length-matched cables for the mixer reference, MIS can be realized with only a moderate sacrifice in SNR[20].

### Nonlinear operation

Nonlinear MIM (NL-MIM) down-converts the microwave reflection at harmonics of the incident microwave frequency and captures local nonlinear response, critical for understanding many systems of practical and fundamental interest, such as doped semiconductors, superconductors, and correlated materials[36–40]. The elimination of cancellation, combined with the adoption of broadband active mixers with a wide range of acceptable reference power, significantly simplifies the construction of NL-MIM.

### Cryogenic operation

The cancellation-free architecture is expected to stay Johnson-noise-limited at cryogenic temperatures because a much lower $P_{in}$ is typically used to prevent local heating or disturbing delicate samples. For example, at 4 K, the combined Johnson and LNA noise will be ∼19 dB lower than that at 295 K, assuming a state-of-the-art cryogenic amplifier. Therefore, for the configuration as in our prototype, the maximum $P_{in}$ for Johnson-noise-limited operation will also be reduced by ∼19 dB, to −29 dBm. However, this is still higher than the −50 to −30 dBm typically used for cryogenic MIM[41,42]. Lower source phase noise or higher probe mechanical resonance will further expand the Johnson-noise-limited regime.

### Higher sensitivity

Higher sensitivity is attainable by further narrowing the measurement bandwidth. Operating in a controlled environment—enclosed, temperature-stabilized, or vacuum—could allow even longer pixel dwell times, albeit with diminishing returns. To further enhance sensitivity one likely needs to combine long pixel dwell time with cryogenic temperature, moderate power, a well-matched probe with high responsivity, and a low-phase-noise source. For example, a combination of −150 dBc/Hz source phase noise figure and a responsivity $\eta = 7.5$ is projected to achieve 0.27 zF sensitivity at 77 K with $P_{in} = −10$ dBm and a pixel dwell time of 160 ms (Fig. 4e).

### Multi-modal integration

The successful utilization of metal-coated monolithic Si probes paves the way to integrate MIM with complementary scanning probe techniques that require (or benefit significantly from) these probes, such as scanning near-field optical microscopy[17], magnetic force microscopy[18], and acoustic force microscopy[19], allowing simultaneous multi-modal characterization of complex materials or devices with multiple degrees of freedom.

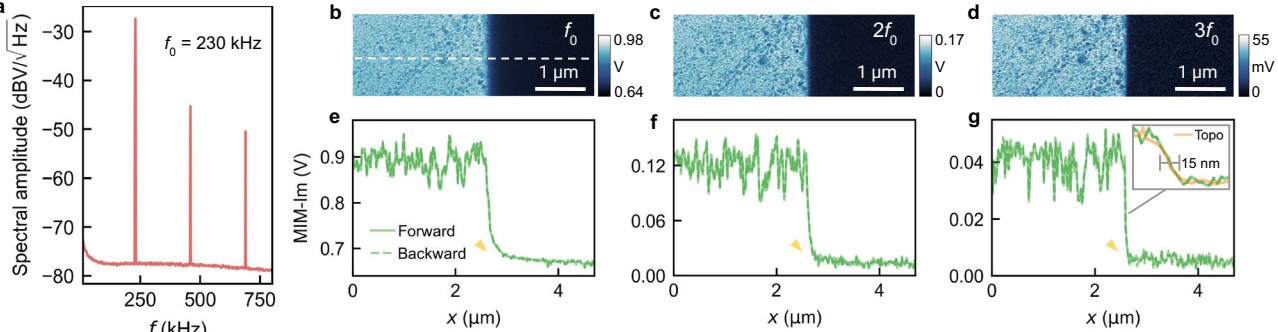

**Fig. 5 | Enhanced spatial resolution through harmonic demodulation. a** MIM spectrum with $P_{in} = -10$ dBm over a wider frequency range, showing dynamic MIM signal at harmonics of $f_0$. **b**–**d** Raw MIM-Im image demodulated at $f_0$, $2f_0$, and $3f_0$, respectively. **e**–**g** Line profiles along the dashed line in the images with a zoomed-in view across the Al/SiO$_2$ edge for $3f_0$ accompanied by the corresponding topography. The yellow arrows indicate the disappearance of the "halo" artifact. Source data are provided as a Source Data file.

## Broader adoption and applications
Our approach significantly lowers the technical and economic barriers to high-performance MIM. Detailed designs and resources are available in Methods and our repository[43], which we hope will facilitate a broad uptake of MIM in the condensed matter physics, microelectronics, and material sciences community. Moreover, since gold-coated silicon probes are widely used to study biological and chemical samples in complex environments[44], our demonstration extends MIM's applicability to these domains, opening avenues for insights into a plethora of topics[22,45,46].

## Methods
### MIM electronics and probes
Here we provide the full list of components for our prototype. The microwave source is Signal Hound VSG60A and the LNA is Lotus LNA700M6G2S. The optional microwave spectrum analyzer for monitoring the reflected microwave is Signal Hound SA124B. The directional couplers are Mini-Circuits ZUDC20-0283-S+. The splitter is Mini-Circuits ZFSC-2-10G+. The phase shifter is PNR P1407. We use Moku: Go as LIA and baseband spectrum analyzer. The monolithic Si cantilever probe with an overall gold coating is OPUS 160AC-GG. The SMC probe is Rocky Mountain Nanotechnology 25PtIr200B-H. The SL probe is PrimeNano M300S.

The design and Gerber files of the impedance matching and mixer-INA PCB can be found in our repository[43]. We use a half-wave resonator made of a 100 mm-long RG-178 coax that gives rise to matched frequencies 1 GHz apart. The coax is wire-bonded to the probes on one end and connected to the MIM electronics via a 0.2 pF series capacitor (Johanson S201TL) on the other end. The active mixer is Analog Devices ADL5380 with a 0.5–6 GHz bandwidth. The INAs are Analog Devices AD8428 with a fixed 2000× voltage gain. We limit the INA bandwidth to 900 kHz.

In addition to the benefits described in the main text, our design also eliminates the need for a tunable attenuator to control $P_{in}$ and a low-phase-noise amplifier for driving the mixer reference port. The active mixer has a virtually constant conversion gain over a very wide range of reference power, so we can change $P_{in}$ simply by changing the source output power through software (Fig. 1a).

### Scanning parameters
We used a Park NX7 atomic force microscope (AFM) to perform the scans. We used a state-of-the-art PrimeNano ScanWave Pro as the commercial MIM electronics to benchmark against. We set the time constant of the LIA to be 1/4 of the pixel dwell time to ensure sufficient settling between pixels. Unless otherwise noted, we used a $P_{in}$ of

−10 dBm at 3.2 GHz and a tapping amplitude of 20 nm. All scans were performed under ambient conditions at 295 K with no enclosure.

### Feasibility of metal-coated Si probes
We model the 160AC-GG probe as the probe body in series with the cantilever. In both regions, the resistance from the 70 nm gold coating ($2.44 \times 10^{-8}$ Ω·m, skin depth 1.3 $\mu$m at 3.2 GHz) is in parallel with the highly-doped silicon substrate ($2.5 \times 10^{-4}$ Ω·m, skin depth 0.14 mm). The silicon contribution is small at GHz frequencies due to the skin effect, so can be ignored here. We estimate 0.4 Ω across the probe body, 0.7 Ω across the cantilever, and less than 1 Ω from the tip pyramid, adding up to ~2 Ω, which is confirmed by our measurements. This figure is moderate compared with the 0.2 Ω of SMC probes and ~5 Ω of SL probes[14].

On the other hand, although Si probes have relatively large unshielded areas, the self-capacitance with a distant ground is estimated and measured (at 100 kHz) to be smaller than 0.2 pF, larger than the ~0.03 pF of SMC or TF probes but less than the ~1 pF of SL probes.

Moreover, with a strategically placed ground plane, much of the capacitance can be compensated by self-inductance to achieve a nearly resistive characteristic impedance close to 50 Ω. To this end, we separated the probe from a ground plane with a 0.5 mm thick fused silica ($\epsilon = 3.8$), forming a microstrip waveguide. The characteristic impedance of this microstrip is estimated to be 44 Ω, close to the system impedance of 50 Ω. This effectively makes the probe body a short extension of the half-wave resonator coax, instead of a large capacitive load. The single-pass attenuation of this microstrip section is estimated to be 0.1 dB at 3.2 GHz from the resistance values above, smaller than the ~0.25 dB from the RG-178 half-wave resonator coax.

### SNR characterization
We measure SNR using two complementary approaches: from MIM-Im images and spectrum analyzer measurements. In the first approach, the signal is the contrast between the average MIM-Im values of Al and SiO$_2$ regions in MIM images (Supplementary Fig. 3a), and the noise is the standard deviation of MIM signal within a $10 \times 10$-pixel feature-less SiO$_2$ region. In the second method, we use a spectrum analyzer to directly measure the spectra of the MIM-Im channel with the resolution bandwidth (RBW) matching the lock-in time constant. We take the difference in the $f_0$ peak amplitude when the tip is on Al or SiO$_2$ as the signal, and the baseline amplitude slightly away from $f_0$ when the tip is on SiO$_2$ as the noise. We then multiply the SNR extracted from the spectrum analyzer by $2/\sqrt{\pi}$ to account for the conversion factor of the LIA. These two methods yield consistent results (Fig. 3d of the main text).

## Signal versus vibration amplitude

The dynamic MIM signal also depends on the cantilever vibration amplitude. Supplementary Fig. 3b shows an example, where the contrast between Al and $SiO_2$ first increases then saturates above ~17.5 nm peak-to-peak amplitude, while the raw signals keep increasing. This trend can be understood from the fact that although the MIM versus tip-sample distance curves are monotonic—hence large vibration amplitude always gives rise to higher dynamic-mode signal, the curves for different materials converge when the tip is far away from the sample, hence the saturation in contrast. We chose 20 nm for an optimal trade-off between MIM signal and topography performance.

## Sensitivity estimation

To convert the total in-band noise level to a capacitance sensitivity, we used the formula[20]

$$V_{MIM} = -G\frac{\Delta Y}{2Y_0}\eta^2 V_{in}, \tag{1}$$

where $G$ is the total gain, $\Delta Y$ is the admittance variation signal, $\eta = 7.5$ is the measured circuit responsivity, $Y_0 = 1/(50\,\Omega) = 0.02$ S is the system characteristic admittance, and $V_{in} = 0.1$ V is the incident microwave voltage for $P_{in} = -10$ dBm. To measure $\eta$, we compared our MIM signal contrast between Al and $SiO_2$ to that from a reference system with a $50\,\Omega$ shunt resistor impedance matching which has an $\eta$ of 1. Setting the MIM root-mean-square (RMS) signal value $V_{MIM}$ to be equal to the total in-band rms noise, we can obtain the corresponding $\Delta Y$ and then estimate the capacitive sensitivity $\Delta C = \Delta Y/(2\pi f)$, where $f = 3.2$ GHz.

## Encapsulated graphene sample fabrication

The sample was fabricated using a dry-transfer[47] method. First, graphene and hBN flakes were exfoliated using tapes onto Si substrates covered by 90 nm thermal oxide. Then, a stamp made of glycol-modified polyethylene terephthalate (PETG) was used to pick up the top hBN, graphene, and bottom hBN in sequence around 70 °C. The stack including the stamp was subsequently released onto a Z-cut quartz substrate at 100 °C and soaked in chloroform overnight to dissolve the PETG. Finally, the sample was washed in chloroform, acetone, isopropanol, deionized water, and dried under nitrogen flow.

## Reporting summary

Further information on research design is available in the Nature Portfolio Reporting Summary linked to this article.

# Data availability

All other data that support the findings of this study are available from the corresponding author upon request. Source data are provided with this paper.

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

## Acknowledgements

Laboratory Directed Research and Development Program of Lawrence Berkeley National Laboratory under U.S. Department of Energy Contract No. DE-AC02-05CH11231: E.Y.M. U.S. Department of Energy, Office of Science, Office of Basic Energy Sciences, Materials Sciences and Engineering Division under contract no. DE-AC02-05-CH11231 (van der Waals heterostructures programme, KCWF16): F.W. Kavli ENSI Heising–Simons Junior Fellowship: S.-D.C.

## Author contributions

J.-Y.S. and E.Y.M. conceived the project. J.-Y.S., N.M., and E.Y.M. constructed the MIM setup and performed the experiments. S.-D.C. and F.W. fabricated the 2D material sample. J.-Y.S. and E.Y.M. wrote the manuscript with inputs from all authors.

## Competing interests

The authors declare no competing interests.
