## [Peer Review File · Nature Communications]

Johnson-noise-limited cancellation-free microwave impedance microscopy with monolithic silicon cantilever probesREVIEWERS' COMMENTS:

Reviewer #1 (Remarks to the Author):

Key Results:

The work by Shan et Al. combines many different techniques to improve the performance of a microwave impedance microscopy system. The listed improvements include:

- "thermally-limited" operation
- cancellation-free architecture of the electronic system, and
- "unprecedented" sensitivity of ~ 2 zF

The work describes the utility of monolithic Silicon probes and provides data on thermally-limited performance, zeptoFarad sensitivity and the use of higher harmonic content for obtaining sharper images.

An important omission in the discussion/references of the manuscript was earlier work from early-mid 2000s that used a similar architecture for zeptoFarad scale capacitive sensing with scanning probe systems. These systems combined microwave resonators with contact probe microscopy as their aim was to measure dopant profiles, but otherwise reached similar level of performance with a similar architecture:

A. Tran, T., et al. "'Zeptofarad'" (10– 21 F) resolution capacitance sensor for scanning capacitance microscopy." *Review of Scientific Instruments* 72.6 (2001): 2618-2623.

- Uses a cantilever coated with Tungsten carbide (i.e. conductive) which is connected to a microwave resonator to obtain close to 1 zF per sqrt Hz capacitive noise level at 90 kHz. This work uses slope detection.

Dongmo, H., Hammond, P., & Weaver, J. (2007, November). Sub-Zeptofarad Sensitivity Scanning Capacitance Microscopy Sensor. In *INTERNATIONAL SYMPOSIUM FOR TESTING AND FAILURE ANALYSIS* (Vol. 33, p. 56)

- Uses a metal coated cantilever coupled to a microwave resonator; uses homodyne and

phase sensitive detection, so the architecture is very similar to the current work.

The absence of these references, and the fact that zeptofarad capacitive detection was demonstrated a while ago with similar architectures bring up questions regarding the suitability of the work for Nature Communications.

Validity:

1. In regards to thermally-limited operation and the mechanical resonance peaks above 100kHz in Figure 3a: are they obtained while the mechanical vibration amplitude was 20 nm, as implied in the "Methods/Scanning Parameters" section ?

If that is the case, then the claim of "thermally-limited" operation should be revisited. In the context of mechanical systems, reaching thermal-limit requires the observation of the thermomechanical peak of the resonator (i.e. the response of an undriven mechanical resonator).

Is the thermomechanical peak of the Silicon cantilever observable with this system?
Otherwise the claim of "thermally-limited" does not seem appropriate.

The flat and drive-independent noise-floor could be coming from some thermal process, but if that thermal process is not the mechanical motion of the cantilever, then it does not seem to be of much relevance for the measurements. Indeed the improvement on Figure 3d does not seem very significant over a commercial system which was presumable not thermally-limited.

2. It is not clear if the measurements in Figure 4 was tried with the commercial system as well. If so, how does the results look? Does the new method presented here make a significant difference for this particular application?

3. Also, for Figure 4d: the drift-free operation of the technique is attained to the cancellation-free operation of the sensor (e..g. self-referenced homodyne). While the sensor receives the benefits of a drift-free architecture, the sample itself was left open to

environment. It is unclear how the temperature and humidity variations of the sample affect the image quality.

Significance:

Observing the noise spectral density and SNR graph of Figure 3c and 3d, the improvement over a commercial system looks modest. For a system to make a significant improvement, an order-of-magnitude (i.e. 10x) advance is typically expected.

Concerns about zeptoFarad detection and thermally-limited performance were discussed above.

So in summary; the work is valuable on various dimensions, however it does not appear to demonstrate a significant enough breakthrough, at least in its current form, for publication in Nature Communications.

In addition to clarifying the thermal-noise limited operation (by demonstrating the thermomechanical peak of the cantilever), and discussing the zeptoFarad results from scanning capacitance microscopy literature where similar sensor architecture was used, a compelling practical demonstration (e.g a 10x improvement over state of the art) is needed to improve the paper.

Minor Point:

1. In Figure 4e, for the dashed line, " theoretical estimates of ... a finer measurement at 77 K" is slightly confusing. Please consider replacing the word "measurement".

Reviewer #2 (Remarks to the Author):

The manuscript by Shan et al presents a new and much simplified method for implementing a scanning microwave impedance microscope (MIM) commonly used in many fields of science to image the complex permittivity response of materials on the nanoscale, at GHz frequencies. The advances presented by the authors address both simplicity in probe

fabrication by the use of monolithic silicon probes rather than specialised RF probes requiring complex manufacture. This yields an improved resolution and reliability due to the much more well-defined silicon tip. The added capacitive crosstalk that comes with this is addressed by the use of an alternative microwave readout scheme. The authors claim that as a result, this is the first MIM which achieves readout performance that is thermally limited. Further improvements can thus be achieved by utilising cold amplifiers, low noise amplifiers. The paper is well written, with sufficient level of detail, and results are clearly presented. The scientific advances are appropriate for Nature communications; however, I have a few questions for clarification before I could recommend the paper for publication in Nature Communications.

Detailed comments:

1. What is the solid line in figure 3d?
2. What was the noise temperature/figure of the amplifier used? How much better performance could be expected using a state of the art cryogenic amplifier with a noise temperature of 1-2K, and how much of the thermal noise then instead originates from the probe?
3. Comparing the 'conventional' and 'this work' in figure 3c, they both level off to a constant slope at low powers, albeit with a small (2dB) difference in amplitude. To me it looks like the 'conventional' system is similarly thermally limited, and cooling that down will also increase its performance. Why is the -186 dBV/rtHz level the thermal limit and the other one not?
4. As the noise is white, it would be convenient to state the noise performance in e.g. mV/rtHz in relation to e.g. figure 4e, and in relation to the quoted sensitivity (zF/rtHz).

Response to Reviewers

We would like to thank the Reviewers for their careful reading of our manuscript "Thermally-limited cancellation-free microwave impedance microscopy with monolithic silicon cantilever probes" (NCOMMS-23-56365) and for their helpful comments. Reviewer #1 wrote that "*The work is valuable in various dimensions*" but had concerns on our terminology and advancements over prior work. Reviewer #2 remarked that "*The paper is well written, with sufficient level of detail, and results are clearly presented. The scientific advances are appropriate for Nature Communications.*" Both had technical questions. Below we address these concerns and questions in detail.

Reviewer #1:

1) *An important omission in the discussion/references of the manuscript was earlier work from early-mid 2000s that used a similar architecture for zeptoFarad scale capacitive sensing with scanning probe systems. These systems combined microwave resonators with contact probe microscopy as their aim was to measure dopant profiles, but otherwise reached similar level of performance with a similar architecture:*

A. Tran, T., et al. "Zeptofarad" (10– 21 F) resolution capacitance sensor for scanning capacitance microscopy." Review of Scientific Instruments 72.6 (2001): 2618-2623. - Uses a cantilever coated with Tungsten carbide (i.e. conductive) which is connected to a microwave resonator to obtain close to 1 zF per sqrt Hz capacitive noise level at 90 kHz. This work uses slope detection.

Dongmo, H., Hammond, P., & Weaver, J. (2007, November). Sub-Zeptofarad Sensitivity Scanning Capacitance Microscopy Sensor. In INTERNATIONAL SYMPOSIUM FOR TESTING AND FAILURE ANALYSIS (Vol. 33, p. 56) - Uses a metal coated cantilever coupled to a microwave resonator; uses homodyne and phase sensitive detection, so the architecture is very similar to the current work.

The absence of these references, and the fact that zeptofarad capacitive detection was demonstrated a while ago with similar architectures bring up questions regarding the suitability of the work for Nature Communications.

We thank the Reviewer for the recommendation to incorporate these important references into our discussion, and the need to explicitly contrast MIM with scanning capacitance microscopy (SCM). Firstly, it is crucial to understand that MIM and SCM cater to distinctly different research needs and material applications. Unlike SCM, which necessitates extensive sample preparation and the establishment of ohmic electrical connections – thereby limiting its applicability primarily to semiconductor device studies – MIM offers a versatile approach suitable for a broad spectrum of materials and structures with minimal preparatory requirements. Additionally, SCM's capability is largely confined to the measurement of real capacitance values. In contrast, MIM has simultaneous quadrature sensitivity and can measure complex permittivity. Therefore, attaining zF sensitivity in MIM stands as a significant achievement in its own right.

Moreover, our bandwidth-normalized sensitivity is in fact significantly higher than previously reported for SCM. To calculate the capacitance sensitivity from measured noise spectra, one needs to know the voltage amplification factor $\eta = V_{\text{tip}}/V_{\text{in}}$ of the impedance matching network [generally a resonator in SCM, but not necessarily the case for MIM, see, e.g., J. Shan *et al.*, “Universal signal scaling in microwave impedance microscopy,” *Appl. Phys. Lett.* 121, 123507 (2022)]. Originally, we conservatively estimated η to be 4. We have since experimentally measured η to be 7.5 by comparing the MIM contrast between Al and SiO₂ using our system and a reference system with a 50 Ω shunt resistor impedance matching that has $\eta = 1$. Consequently, we report a sensitivity of 0.53 zF at a 4 Hz acquisition bandwidth (Fig. 4g; this is distinct from the ~ 1 MHz baseband bandwidth), leading to a bandwidth-normalized sensitivity of 0.26 zF/rtHz. This represents a significant enhancement over the ~ 1 zF/rtHz reported in previous SCM work.

To clarify the points above and improve transparency and context, we have incorporated the SCM references and modified the texts: In the “Methods” section we have added: “To measure η , we compared our MIM signal contrast between Al and SiO₂ to that from a reference system with a 50 Ω shunt resistor impedance matching which has an η of 1.”; in the “Sub-zeptofarad sensitivity” section, we used the measured η and added a comparison with SCM: “...or a bandwidth-normalized sensitivity of 0.26 zF/rtHz, representing a 3-4x improvement over previously reported figures in the related scanning capacitance microscopy (SCM) [30,31]. We note that MIM differs significantly from SCM due to its simultaneous quadrature sensitivity and minimal sample preparation requirements.”

2) In regards to thermally-limited operation and the mechanical resonance peaks above 100kHz in Figure 3a: are they obtained while the mechanical vibration amplitude was 20 nm, as implied in the “Methods/Scanning Parameters” section ?

If that is the case, then the claim of “thermally-limited” operation should be revisited. In the context of mechanical systems, reaching thermal-limit requires the observation of the thermomechanical peak of the resonator (i.e. the response of an undriven mechanical resonator).

*Is the thermomechanical peak of the Silicon cantilever observable with this system?
Otherwise the claim of “thermally-limited” does not seem appropriate.*

The flat and drive-independent noise-floor could be coming from some thermal process, but if that thermal process is not the mechanical motion of the cantilever, then it does not seem to be of much relevance for the measurements.

We thank the Reviewer for pointing out the need for more precise language regarding “thermal noise”. Our reference to “thermally-limited performance” is not about the cantilever *mechanical* motion, but the Johnson-Nyquist noise in our *electrical* circuit (as the Reviewer suggested in the last paragraph). This performance is substantiated experimentally, as our input-referenced noise floor (Fig. 3c) quantitatively agrees with the Johnson noise plus the LNA noise figure of 0.6 dB. We have changed our title to “**Johnson-noise-limited**” to clarify this crucial point. However, we respectfully disagree

with the notion that this electrical noise “does not seem to be of much relevance for the measurements”, because we showed that it is the dominant noise source for MIM.

With regards to the questions on mechanical vibration amplitude, in both Fig. 3a and Fig. 3b we were operating in tapping mode, where the cantilever was driven and oscillating with an amplitude of 20 nm. We performed additional experiments as the Reviewer suggested, and the thermomechanical peak of the *undriven* silicon cantilever is clearly observed, as expected for such high quality probes (see figure below). Nonetheless, there is little benefit to rely on such thermal motion for topography or MIM measurements vs. the standard tapping mode with a coherently driven cantilever.

3) *It is not clear if the measurements in Figure 4 was tried with the commercial system as well. If so, how does the results look? Does the new method presented here make a significant difference for this particular application?*

We performed the measurements in Fig. 4 using a commercial system with a cancellation circuit. (See the figures below. Left: 6-hour scan with our cancellation-free MIM. Right: 6-hour scan with commercial MIM.) The result is that after ~2 hours, the commercial system drifted out of cancellation and the baseband amplifiers were saturated, yielding a null MIM signal from that point on. This is highly consistent with the conventional rule of thumb that one should limit MIM image acquisition to within an hour or so, even in a tightly controlled lab environment. Our cancellation-free architecture removed this limit, and thus allows much longer pixel dwell times and order-of-magnitude improvements in sensitivity. We note that 6 hours is not the limit here – we expect multi-day measurements to be possible, limited only by the stability of the scanning probe platform and sample degradation, as is the case for scanning tunneling microscopy.

We clarified this in the last paragraph of the “Sub-zeptofarad (zF) sensitivity” section: “Under identical

settings, the commercial system drifted into saturation in less than 2 hours.”

4) *Also, for Figure 4d: the drift-free operation of the technique is attained to the cancellation-free operation of the sensor (e.g. self-referenced homodyne). While the sensor receives the benefits of a drift-free architecture, the sample itself was left open to environment. It is unclear how the temperature and humidity variations of the sample affect the image quality.*

We agree with the Reviewer that the temperature and humidity variations will cause changes in the sample complex permittivity, and for sensitive samples it is beneficial to place the sample inside an enclosed, actively-controlled environment. That being said, we went back and checked the temperature and humidity log of the lab when taking Fig. 4d and 4g, and found that the maximum temperature variation is $\sim 0.5^\circ\text{C}$ and the maximum humidity variation is 3%. Such moderate environmental variation is not expected to deteriorate the image quality appreciably for the graphene/hBN samples we tested.

5) *...Indeed the improvement on Figure 3d does not seem very significant over a commercial system which was presumably not thermally-limited*

...

Observing the noise spectral density and SNR graph of Figure 3c and 3d, the improvement over a commercial system looks modest. For a system to make a significant improvement, an order-of-magnitude (i.e. 10x) advance is typically expected.

While it is true that improvements are modest in the noise spectral density (Fig. 3c) and SNR for fixed pixel dwell time (Fig. 3d), we did achieve a nearly order-of-magnitude enhancement in the ultimate *sensitivity* or ultimate SNR, a more important metric for scanning probes like MIM. We achieved this because in addition to lower noise density, our cancellation-free architecture eliminates drifts and allows significantly extended pixel dwell times and lower *total in-band noise*. We discuss and experimentally demonstrate this point in detail in the “Sub-zerofarad (zF) sensitivity” section (see e.g. Fig. 4e, and response to comment#3 above).

Another significant advance we achieved is a baseband bandwidth of ~ 1 MHz, a threefold increase over the best commercial systems. This enhancement enables us to demodulate at higher harmonics of the cantilever's mechanical resonant frequency, thus significantly improving the *electronic spatial resolution* of MIM – another crucial metric for scanning probe techniques (Fig. 5).

Taking all these advancements into consideration – the Johnson-noise-limited noise floor, the longer pixel dwell times afforded by our cancellation-free design, and the substantial increase in baseband bandwidth – it becomes evident that our system now allows mapping previously inaccessible minuscule complex permittivity variations over nanometer length scales in a wide range range of materials and devices. We trust that this embodies a qualitative leap forward in the performance and versatility of MIM over existing commercial solutions.

We added a discussion in the last paragraph of the “Johnson-noise-limited performance” section: “We note that here the comparison between SNR at different conditions does not take into account the dwell time per pixel, and the removal of the cancellation circuit further allows robust drift-free operation, enabling much longer pixel dwell time and thus higher sensitivity.”

6) *So in summary; the work is valuable on various dimensions, however it does not appear to demonstrate a significant enough breakthrough, at least in its current form, for publication in Nature Communications.*

In addition to clarifying the thermal-noise limited operation (by demonstrating the thermomechanical peak of the cantilever), and discussing the zeptoFarad results from scanning capacitance microscopy literature where similar sensor architecture was used, a compelling practical demonstration (e.g a 10x improvement over state of the art) is needed to improve the paper.

We hope that by clarifying the terminology of “thermal noise”, differentiating MIM from SCM, and demonstrating significant improvements along various dimensions over both SCM and commercial MIM, we have addressed the concerns raised by the Reviewer.

7) *Minor Point:*

1. *In Figure 4e, for the dashed line, "theoretical estimates of... a finer measurement at 77 K" is slightly confusing. Please consider replacing the word "measurement".*

We thank the Reviewer for the careful reading of our manuscript. We changed the word to “proposed measurement.”

Reviewer #2:

1) *What is the solid line in figure 3d?*

We apologize for having the wrong labeling in our previous manuscript. The solid line is a fit of the experimental data (this work, extracted from MIM signal spectra) for $P_{in} < -10$ dBm. In this regime the noise is constant (thermal Johnson noise), and the signal scales with P_{in} , so the SNR scales like $10^{P_{in}(\text{dBm})/20}$ which fits the data well for $P_{in} < -10$ dBm.

We corrected the figure labeling in the main text and clarified this in the caption of Fig. 3.

2) *What was the noise temperature/figure of the amplifier used? How much better performance could be expected using a state of the art cryogenic amplifier with a noise temperature of 1-2K, and how much of the thermal noise then instead originates from the probe?*

The noise figure of our room-temperature LNA is 0.6 dB, corresponding to a noise temperature of 44 K at a reference temperature of 295 K. For a state-of-the-art cryogenic LNA such as the CITLF2 from

Cosmic Microwave Technology (<https://www.cosmicmicrowavetechnology.com/citlf2>), at room-temperature, its noise temperature is in fact higher, at ~ 70 K, so it would not enhance performance; if cooling both the probe and the LNA to 12 K, the CITLF2's 3 K noise temperature or 1 dB noise figure means that the fractional contribution of LNA noise to the total noise will be similar to the case at room temperature with our LNA. Nonetheless, the absolute total noise will be lower by ~ 14 dB due to the lower temperature.

We clarified this when estimating the noise floor at low temperatures, in the “Cryogenic operation” subsection of the Discussion section: “..., assuming a state-of-the-art cryogenic amplifier.”

3) Comparing the ‘conventional’ and ‘this work’ in figure 3c, they both level off to a constant slope at low powers, albeit with a small (2dB) difference in amplitude. To me it looks like the ‘conventional’ system is similarly thermally limited, and cooling that down will also increase its performance. Why is the -186 dBV/rtHz level the thermal limit and the other one not?

We calculated the *input-referenced* Johnson noise spectral density as $k_B T$, where k_B is the Boltzmann constant and T is temperature, resulting in -173.8 dBm/Hz, or -186.8 dBV/rtHz for 50 Ω . Adding a 0.6 dB noise figure from the LNA, we reach -186.2 dBV/rtHz – this is thus the thermal noise floor, ignoring the small noise contribution from downstream amplifiers. We then measure the noise spectra of both systems, and subtract the independently calibrated total system gain. For our home-built system, the number comes out to be ~ -186 dBV/rtHz whereas that for the commercial system is ~ 2 dB higher. Therefore, we conclude that there are additional broadband noise sources in the conventional system, such as the noise from the cancellation line, a much higher LNA noise figure, or excessive noise from downstream components coupled with a low-gain LNA. We agree with the Reviewer that cooling the commercial system down will increase its performance due to the reduction of the thermal Johnson noise. However, the additional noise – possibly temperature-independent – likely cannot be removed, and may become increasingly detrimental at lower temperatures.

We clarified this in the first two paragraphs in the “Johnson-noise-limited performance” section: “The input-referenced Johnson noise density is $k_B T$, where k_B is the Boltzmann constant and T is temperature, ... , is -186.8 dBV/rtHz for 50 Ω .”

4. As the noise is white, it would be convenient to state the noise performance in e.g. mV/rtHz in relation to e.g. figure 4e, and in relation to the quoted sensitivity (zF/rtHz).

We thank the Reviewer for the suggestion. The rms noise voltage density is 0.14 mV/rtHz, for the quoted bandwidth-normalized sensitivity of 0.26 zF/rtHz. We have added this in the caption of Fig. 4e.

REVIEWER COMMENTS

Reviewer #1 (Remarks to the Author):

The authors have provided new data sets and made careful edits on the paper based on the comments from both reviewers.

1. While there are differences between SCM and MIM, this paper's contribution is not along the dimension of these differences. The authors supplied a balanced discussion on this point. So no further changes or comments are needed here.

2. For point 2 (thermal vs Johnson), the authors have clarified the situation at several parts of the paper and showed a readout of thermomechanical peak. The visibility of the (undriven) thermomechanical peak is an important noise consideration, regardless of the actual amplitude of operation. It shows that the mechanical domain signal is not dominated by other sources.

2A. Authors mention on their response that "...this electrical noise ... is the dominant noise source for MIM."

But then from the noise comparison with the commercial system, it looks like the difference is only 2 dB. From the data supplied to Figure 4d (drift of the commercial system), I suggest the authors look at Allan Deviation at relatively long time scales, and report that graph to show their drift-free performance level in a more quantitative manner that relates to phase noise measurements.

2B. One peculiar region in Figure 3c is the high Pin values (such as Pin=-5dB). Here the commercial system has lower noise. Could the authors mention if this would form a limitation for the technique presented, and the potential reason for the cross-over of the noise levels.

Point 3 and Point 4 are addressed (but please check the comment on Allan Deviation for drift).

Point 5: I understand the author's point. I suggest that in the SNR graph of Figure 4d, they

also make it clear that the performance level of “commercial system with Silicon probe” relates to the adoption of the Silicon probe in this work, and not a standard feature of a commercial system, to clarify the contribution of the paper.

Overall, the authors have proposed and implemented a practical way with improvements on multiple fronts, and the paper can be re-evaluated after several points mentioned above.

Reviewer #2 (Remarks to the Author):

The authors have answered all my previous queries satisfactory, and improved the manuscript accordingly.

Response to Reviewers

We would like to thank the Reviewers for their careful reading of our revised manuscript "Johnson-noise-limited cancellation-free microwave impedance microscopy with monolithic silicon cantilever probes" (NCOMMS-23-56365A-Z) and for their helpful additional comments. Reviewer #2 has no more comments. Below we address the remaining comments from Reviewer #1.

Reviewer #1:

1) *For point 2 (thermal vs Johnson), the authors have clarified the situation at several parts of the paper and showed a readout of thermomechanical peak. The visibility of the (undriven) thermomechanical peak is an important noise consideration, regardless of the actual amplitude of operation. It shows that the mechanical domain signal is not dominated by other sources.*

We thank the Reviewer for pointing out the importance of the thermomechanical noise. In the last paragraph of the "Monolithic silicon probes for microwave" section we have added: "This allows us to unlock key benefits of Si probes, such as ...thermally-limited mechanical noise, ..."

2) *Authors mention in their response that "...this electrical noise ... is the dominant noise source for MIM." But then from the noise comparison with the commercial system, it looks like the difference is only 2 dB. From the data supplied to Figure 4d (drift of the commercial system), I suggest the authors look at Allan Deviation at relatively long time scales, and report that graph to show their drift-free performance level in a more quantitative manner that relates to phase noise measurements.*

We thank the Reviewer for pointing out the need for more quantitative analysis of our drift-free performance. We conducted further experiments where we continuously measured the MIM-Im signal on Al for 6 hours. Measurements were taken at the same spot with intervals of 160 ms. We compared the performance between our cancellation-free system and the commercial system.

We have reported the new data as the Extended Fig. 3 and cited it in the main text (also see below). In the time domain, the commercial system quickly drifted into saturation after ~1 h, while our cancellation-free system maintained drift-free. Examining the Allan deviation (Extended Fig. 3b), we arrived at the same conclusion: the Allan deviation of the commercial system at long time scales is more than an order of magnitude larger than that of our cancellation-free system.

3) *One peculiar region in Figure 3c is the high P_{in} values (such as $P_{in}=-5\text{dB}$). Here the commercial system has lower noise. Could the authors mention if this would form a limitation for the technique presented, and the potential reason for the cross-over of the noise levels.*

At high microwave power, the system is no longer limited by the Johnson noise, but by the phase noise of the microwave source, which can be partially mitigated by the cancellation circuit. This is the reason why at high microwave power the commercial system has a lower overall noise level. However, we note that practically this is not a limitation, because a routinely used $P_{in} < -10\text{ dB}$ is enough for a good signal-to-noise ratio, and very often one would like to avoid high P_{in} in order to prevent perturbation to the sample.

We clarified this in the third paragraph of the “Johnson-noise-limited noise floor” section: “... cancellation can mitigate phase noise at higher power levels (e.g., for \$P_{in}>-6\text{ dBm}\$ in Fig. 3c), ...”

4) *I understand the author’s point. I suggest that in the SNR graph of Figure 3d, they also make it clear that the performance level of “commercial system with Silicon probe” relates to the adoption of the Silicon probe in this work, and not a standard feature of a commercial system, to clarify the contribution of the paper.*

We clarified this in the caption of Fig. 3d: “We note that Si probes are not a standard feature of any commercial system.”

REVIEWERS' COMMENTS

Reviewer #1 (Remarks to the Author):

I think the manuscript is suitable for publication after the changes the authors conducted.